# Deep-Learning-Based Computer-Aided Grading of Cervical Spinal Stenosis from MR Images: Accuracy and Clinical Alignment

**DOI:** 10.3390/bioengineering12060604

**Published:** 2025-06-01

**Authors:** Zhiling Wang, Xinquan Chen, Bin Liu, Jinjin Hai, Kai Qiao, Zhen Yuan, Lianjun Yang, Bin Yan, Zhihai Su, Hai Lu

**Affiliations:** 1Fifth Affiliated Hospital of Sun Yat-sen University, Department of Spinal Surgery, 52 East Meihua Rd, Xiangzhou, Zhuhai 519000, China; 2Department of Orthopedics, The First Affiliated Hospital of Jinan University, Guangzhou 510630, China; 3Henan Key Laboratory of Imaging and Intelligence Processing, Information Engineering University, Zhengzhou 450001, China; 4Faculty of Health Sciences, University of Macau, Taipa, Macau 999078, China

**Keywords:** deep learning, cervical spinal stenosis, classification, target detection, automatic measurement

## Abstract

**Objective:** This study aims to apply different deep learning convolutional neural network algorithms to assess the grading of cervical spinal stenosis and to evaluate their consistency with clinician grading results as well as clinical manifestations of patients. **Methods:** We retrospectively enrolled 954 patients with cervical spine magnetic resonance imaging (MRI) data and medical records from the Fifth Affiliated Hospital of Sun-Yat Sen University. The Kang grading method for sagittal MR images of the cervical spine and the spinal cord compression ratio for horizontal MR images of the cervical spine were adopted for cervical spinal canal stenosis grading. The collected data were randomly divided into training/validation and test sets. The training/validation sets were processed by various image preprocessing and annotation methods, in which deep learning convolutional networks, including classification, target detection, and key point localization models, were applied. The predictive grading of the test set by the model was finally contrasted with the grading results of the clinicians, and correlation analysis was performed with the clinical manifestations of the patients. **Result:** The EfficientNet_B5 model achieved a five-fold cross-validated accuracy of 79.45% and near-perfect agreement with clinician grading on the test set (*κ*= 0.848, 0.822), surpassing resident–clinician consistency (*κ* = 0.732, 0.702). The model-derived compression ratio (0.45 ± 0.07) did not differ significantly from manual measurements (0.46 ± 0.07). Correlation analysis showed moderate associations between model outputs and clinical symptoms: EfficientNet_B5 grades (*r* = 0.526) were comparable to clinician assessments (*r* = 0.517, 0.503) and higher than those of residents (*r* = 0.457, 0.448). **Conclusion:** CNN models demonstrate strong performance in the objective, consistent, and efficient grading of cervical spinal stenosis severity, offering potential clinical value in automated diagnostic support.

## 1. Introduction

Degenerative cervical myelopathy (DCM), formerly known as cervical spondylotic myelopathy (CSM), is the most common cause of cervical spinal cord dysfunction [1]. Characterized by intervertebral disc herniation, osteophyte formation, and the ossification of the posterior longitudinal ligament, cervical spinal canal stenosis can lead to spinal cord compression and subsequent myelopathic symptoms in DCM, including limb pain, numbness, poor coordination, imbalance, and bladder dysfunction [2,3]. With global population aging, the prevalence of DCM is increasing worldwide. According to the published data in the United States, DCM is observed in approximately 13% of the elderly population (≥65 years old) in 2010, which is projected to rise to 22% by 2050 [2]. The data in other developed or developing countries, including China, showed similar trends [4].

The diagnosis of DCM should be based on the comprehensive evaluation of clinical manifestations and imaging findings. However, it is reported that over 70% of elderly individuals (≥65 years old) exhibit radiological spinal cord compression, yet only approximately 25% among them develop symptomatic myelopathy. That “clinical–radiological mismatch” might be attributed to the spinal cord’s resilience to chronic compression or subjective differences among individuals [5]. Notably, imaging evidence alone, without typical clinical symptoms, is insufficient for the diagnosis of DCM and thus defined as preclinical spondylotic cervical cord compression (PSCCC), for which 8% of patients progressed to DCM within one year and 22% developed DCM during an observation period of 2–12 years [6]. As for the management of PSCCC, surgery and conservative treatment are alternative options and may depend on the presence of radicular symptoms and electrophysiological abnormalities [2]. In this way, identifying the risk of progression to DCM in PSCCC patients is an essential process for stratification, necessitating a more refined classification system for cervical spinal canal stenosis and spinal cord compression.

Magnetic resonance imaging (MRI), especially sagittal T2-weighted imaging, plays a crucial role in detecting cervical spinal canal stenosis, spinal cord compression, and lateral recess and foraminal stenoses, which are prerequisites for the diagnosis of DCM [7]. Current research suggests that the presence of radicular symptoms, high signal intensity on T2-weighted MRI, and electrophysiological abnormalities are significant predictors of PSCCC progression to DCM [6,8]. Based on sagittal T2-weighted MRI, the Kang grading system incorporates high signal intensity of the spinal cord into its classification and has been reported to correlate with radicular symptoms so as to provide more precise risk assessments for DCM progression [9,10]. However, proficiency in Kang grading requires extensive clinical experience, and manual grading remains subjective, time-consuming, and prone to errors, particularly when evaluating a large number of images. Moreover, the interobserver agreement in Kang grading system cannot be overlooked [9,11]. Accordingly, the Park-modified Kang grading system was proposed, but it largely relies on experienced neuroradiologists and musculoskeletal radiologists [10]. These limitations restrict the clinical application of the Kang grading system and its potential role in risk stratification.

Recently, deep learning-based approaches have demonstrated superior performance in various medical imaging analysis tasks, such as detecting pulmonary nodules and intracranial hemorrhages on CT scans [12,13]. Deep learning has also been extensively applied in spinal imaging, including automated segmentation [14,15] and grading of lumbar disc herniation and central canal stenosis [16,17,18]. However, most studies have focused on the lumbar spine, and research on deep-learning-based image analysis for cervical spinal cord compression remains limited [19,20]. Moreover, no studies have combined deep-learning-based Kang grading with clinical symptoms.

Therefore, this study aims to develop a deep learning model capable of performing Kang grading on sagittal T2-weighted MRI. The deep-learning-based grading tool developed in this study is expected to assist clinicians in providing objective, reliable, and rapid assessments for evaluating the severity of degenerative cervical myelopathy (DCM) and estimating the risk of progression from PSCCC to DCM. This approach holds particular promise for primary healthcare centers, where it can enhance the role of imaging in clinical decision-making, and offers a practical and efficient grading solution for future research into the clinical relevance and prognostic implications of cervical stenosis classification.

## 2. Materials and Methods

### 2.1. Data Collection

This single-center retrospective study was approved by the Institutional Review Board (IRB) of the Fifth Affiliated Hospital of Sun Yat-sen University. A total of 1000 patients (including inpatients and outpatients) who were diagnosed with cervical spondylosis and underwent cervical spine MRI between January 2018 and March 2021 in the Fifth Affiliated Hospital of Sun Yat-sen University were included in our study. Exclusion criteria included (1) patients without sagittal T2-weighted spin-echo (SE) images; (2) severe artifacts on MRI; (3) presence of surgical implants; and (4) presence of peripheral nerve compression. The interval between clinical symptom evaluation and MRI image review did not exceed one month for any patient.

A total of 1000 cervical MRI studies were initially retrieved according to the predetermined sample size calculation. Based on the exclusion criteria, 12 cases were excluded due to poor image quality, 24 due to artifacts from surgical implants, and 10 due to incomplete imaging sequences. Ultimately, 954 eligible cases with both sagittal and axial T2-weighted MRI were included and saved in DICOM format, meeting the requirement of at least 900 cases as estimated by sample size calculation. The dataset was randomly divided into a training/validation set and a test set in an 8:2 ratio. The training/validation set consisted of 764 cases, while the test set comprised 190 cases, each including both sagittal and axial T2-weighted MRI images. MRI examination of the cervical spine was performed using a 3.0 T, 8-channel coil MRI system (Magnetom Verio, Siemens Health Care, Erlangen, Germany).

### 2.2. Image Preprocessing

Data collectors used MicroDicom 2022.1 to select mid-sagittal T2-weighted images for each patient and converted them to JPG format. Labeling was performed using Lableme, where the region of interest (ROI) was annotated. The ROI was defined as a rectangular region with the superior and inferior boundaries at the upper and lower edges of the adjacent vertebral bodies at the site of the most severe spinal canal stenosis. The anterior and posterior boundaries were set at the anterior vertebral edge and the posterior margin of the spinous process, respectively. The images were cropped based on the ROI to serve as the grading image for each patient. Each patient had only one representative preprocessed image.

### 2.3. Dataset Annotation

The preprocessed images were randomly divided into training/validation and test sets at an 8:2 ratio. For the training/validation set, two junior resident physicians with less than three years of clinical experience (reader1 and reader2) independently assigned Kang grades to the preprocessed images after face-to-face joint interpretation of an additional 50 cases to standardize the Park-modified Kang grading system’s assignment at the beginning of the study (Figure 1). There was no time restriction for grading. The interobserver agreement between the two physicians was assessed using Cohen’s kappa coefficient. The interpretation of Cohen’s kappa coefficient (κ) was as follows: values of 0 < *κ* ≤ 0.20 indicated slight agreement, 0.21 < *κ* ≤ 0.40 indicated fair agreement, 0.41 < *κ* ≤ 0.60 indicated moderate agreement, 0.61 < *κ* ≤ 0.80 indicated substantial agreement, and 0.81 < *κ* ≤ 1.00 indicated almost perfect agreement.

For the test set, two senior physicians with more than ten years of clinical experience (reader3 and reader4) independently graded the images, and their evaluations served as the reference standard. To compare with the model and assess interobserver variability, the test set was also graded by the two junior physicians (reader1 and reader2).

### 2.4. Model Training

During model training, we explored various commonly used image recognition network architectures through five-fold cross-validation to determine the most suitable framework for our dataset. All models were initialized with weights pre-trained on the ImageNet dataset. The Adam optimizer was employed with an initial learning rate of 0.001, which was adjusted using a cosine annealing decay schedule. A batch size of 16 was applied for all experiments. The cross-entropy loss on the validation set was monitored throughout the training process to evaluate model performance and prevent overfitting. 

Due to the limited amount of training data, particularly for grade 3 images, extensive data augmentation was performed. The images underwent random horizontal flipping, random vertical flipping, random rotation (−10° to 10°), contrast enhancement (1–1.5×), brightness variation (0.7–1.3×), and normalization.

The initial weights of all models were derived from transfer learning using pre-trained ImageNet weights. Transfer learning allows a model to initialize with weights trained on another image classification task rather than random weights, making it easier to adapt since the early layers of a CNN detect fundamental image features such as edges, which are generalizable across different classification tasks. Transfer learning can potentially improve model training and performance on smaller datasets [21].

The model with the highest average accuracy from five-fold cross-validation was selected, which was EfficientNet. EfficientNet is based on a simple yet effective compound scaling method that efficiently scales a baseline ConvNet to meet target resource constraints while maintaining model efficiency. Compared to existing CNN architectures such as AlexNet, ResNet, GoogleNet, and MobileNetV2, EfficientNet achieves higher accuracy and efficiency [22].

EfficientNet includes models from B0 to B7, with parameters ranging from 5.3 M to 66 M. To identify the most suitable model for this dataset, we conducted five-fold cross-validation and ultimately selected EfficientNet-B5, which achieved the highest average accuracy.

The experiments were conducted on a PC equipped with an Intel i7-8700K 3.7 GHz CPU, a GeForce RTX 3070 GPU, and 64 GB of RAM. All algorithms were implemented using TensorFlow.

### 2.5. Model Testing

The best-performing model architecture, EfficientNet-B5, was applied to the test set. The model generated a predicted grade for each preprocessed test image, which was then compared to the reference standard provided by the two senior physicians (reader3 and reader4). Evaluation metrics, including area under the curve (AUC), sensitivity, specificity, and F1-score, were computed.

To compare performance and assess interobserver variability, an agreement analysis was conducted among the model, the two junior physicians (reader1 and reader2), and the two senior physicians (reader3 and reader4) in grading the test set.

### 2.6. Correlation Between Grading and Clinical Radicular Symptoms

To investigate the correlation between grading and clinical radicular symptoms, patients in the test set were screened. Exclusion criteria included outpatients, acute trauma cases, a history of cervical spine surgery, comorbid conditions such as cerebral infarction or other intracranial pathologies, and peripheral neuropathies such as carpal tunnel syndrome. A retrospective review was conducted on the remaining hospitalized patients, including their medical history and physical examinations.

Positive neurological symptoms included documented upper limb sensory abnormalities, weakness, numbness, or radicular pain. Positive neurological signs included positive Lhermitte’s sign, positive Spurling’s test, decreased deep tendon reflexes, and electromyography (EMG) evidence of denervation. The presence of more than one positive neurological symptom combined with more than one neurological sign was considered indicative of clinically relevant radicular symptoms [23].

The correlation between grading by the model, the two junior physicians (reader1 and reader2), and the two senior physicians (reader3 and reader4) and the presence of radicular symptoms was analyzed.

### 2.7. Statistical Analyses

Prior to the initiation of this retrospective study, the minimum required sample size was estimated. According to the previous literature [24,25], the approximate distribution of patients across Kang grading categories for cervical spinal stenosis is 10%, 40%, 30%, and 20%, respectively. To evaluate the agreement between model-predicted grading and clinician assessments, the minimum sample size for the test set was calculated based on Cohen’s kappa statistic using PASS software, yielding a requirement of at least 90 cases.

To enhance the generalizability of the deep learning models, we planned to include both outpatient and inpatient MR imaging data. Given the approximate 1:1 ratio of outpatient to inpatient MR examinations in the hospital’s radiology system, the target sample size for the test set was set at 180 cases.

The dataset was randomly divided into training/validation and test sets at an 8:2 ratio. The training/validation set was used to train the deep learning models, while the test set was reserved for the final performance evaluation. Accordingly, 720 cases were allocated to the training/validation set and 180 to the test set, resulting in a total planned sample size of 900 cases. Considering a potential data loss rate of approximately 10%, a total of 1000 cervical MR imaging cases were planned for collection.

## 3. Results

### 3.1. Demographics

The demographic characteristics of the training/validation and test datasets, including mean age and sex distribution. The mean age was 46.65 ± 12.68 years in the training/validation set and 47.72 ± 11.85 years in the test set, with no statistically significant difference between the two groups (*p* = 0.2937). The proportion of male patients was 40.84% in the training/validation set and 46.32% in the test set, also showing no significant difference (*p* = 0.1712). These results indicate that the randomization of patients into the training/validation and test sets was successful.

### 3.2. Kang Grading Prediction in the Test Set Using EfficientNet

To further validate the performance of the EfficientNet_B5 model, we applied the trained deep learning model to predict Kang grades for the preprocessed T2-weighted sagittal MR images of 190 patients in the test set. Additionally, two resident doctors (Resident 1 and Resident 2), after an interval of one month following their assessment of the training/validation set, independently graded the test set images to serve as comparative references.

As the grading of cervical spinal canal stenosis based on MRI relies on visual reference images and is inherently subjective, lacking a universally accepted gold standard, it was necessary to establish an expert-based relative standard. Therefore, two experienced orthopedic clinicians (Clinician 1 and Clinician 2, each with ≥10 years of clinical experience) independently graded the test set images after a thorough literature review and with reference to standardized grading diagrams. Their assessments served as the relative gold standard for evaluating grading consistency.

The distribution of Kang grades predicted by the two resident doctors, two orthopedic clinicians, and the EfficientNet_B5 model (CNN) in the test set is presented in Figure 1. The overall proportions of each Kang grade in the test set were as follows: Grade 0 accounted for 8–12%, Grade 1 for 34–39%, Grade 2 for 41–47%, and Grade 3 for 9–11%. The distribution patterns of grading results by the residents, clinicians, and the deep learning model were comparable, indicating the model’s grading tendencies closely aligned with those of human raters.

### 3.3. Training/Validation Set

Two junior physicians (reader1 and reader2) independently graded the 764 images in the training/validation set using the Park grading system. Table 1 presents the confusion matrix of Kang grading by reader1 and reader2 for the training/validation set. The interobserver agreement between reader1 and reader2 was calculated as Cohen’s *κ* = 0.678.

### 3.4. Model Training

The training and validation sets were used for model development. The dataset consisted of 764 preprocessed sagittal T2-weighted MR images, with one representative image selected per patient. Five commonly used image classification models were trained and evaluated using five-fold cross-validation to compare their average grading accuracies (Table 2). As shown in Table 2, EfficientNet_B0 achieved the highest average grading accuracy of 77.89% across the five folds.

Considering that the EfficientNet architecture includes multiple configurations with varying depths and widths, we further performed five-fold cross-validation on different EfficientNet variants to identify the optimal model for our dataset. The results indicated that EfficientNet_B5 achieved the highest average grading accuracy of 79.45% on the validation set (Table 3). Consequently, the best-performing EfficientNet_B5 model from cross-validation was selected for subsequent testing on the independent test set.

### 3.5. Model Testing

The Kappa statistics for grading consistency between two residents (Resident 1, Resident 2), two orthopedic clinicians (Clinician 1, Clinician 2), and the EfficientNet_B5 deep learning model (CNN) are summarized in Table 4 and Table 5. Kappa values were interpreted as follows: 0 < *κ* ≤ 0.2 (slight), 0.2 < *κ* ≤ 0.4 (fair), 0.4 < *κ* ≤ 0.6 (moderate), 0.6 < *κ* ≤ 0.8 (substantial), and 0.8 < *κ* ≤ 1 (almost perfect).

The consistency analysis for Kang grading of cervical spinal stenosis revealed that the grading agreement between the two orthopedic clinicians was nearly perfect (*κ* = 0.926), while the residents showed good agreement (*κ* = 0.684). These results align with Lee et al.’s study, which found that two experienced orthopedic radiologists achieved almost perfect agreement (*κ* = 0.912), whereas two non-specialist physicians had good agreement (*κ* = 0.691). Furthermore, trained junior radiologists reached almost perfect consistency (*κ* = 0.890) with specialists.

Although Kang grading shows good consistency among experienced clinicians, it requires a learning curve for less experienced physicians. The CNN model achieved nearly perfect agreement with the orthopedic clinicians (*κ* = 0.848, *κ* = 0.822), outperforming the residents’ consistency with clinicians (*κ* = 0.732, *κ* = 0.702). This suggests that despite being trained by residents, the CNN model’s performance on the test set exceeds that of less experienced physicians, demonstrating the feasibility of deep learning models in cervical spine grading. This also highlights the potential of deep learning to enhance Kang grading’s clinical application and offers a reliable, efficient tool for large-scale clinical data analysis.

The Kang grading results for cervical spinal stenosis in 104 hospitalized patients from the test set, as assessed by two residents (Resident 1 and Resident 2), two attending orthopedic clinicians (Clinician 1 and Clinician 2), and the convolutional neural network (CNN) model, are summarized in Table 6. The distribution of Kang grades assigned by all raters and the CNN model was as follows: Grade 0 accounted for 4–8%, Grade 1 for 29–36%, Grade 2 for 46–51%, and Grade 3 for 12–14%. Compared with the overall grade distribution in the entire test set, the proportion of Grades 0 and 1 was lower in hospitalized patients, while the proportions of Grades 2 and 3 were higher. This trend is consistent with clinical expectations, as patients with more severe clinical symptoms are more likely to be hospitalized for further management, resulting in a decreased proportion of normal cervical spines and an increased proportion of severe stenosis cases.

In addition, the presence of positive clinical signs of cervical spinal stenosis was analyzed across different Kang grades. None of the patients classified as Grade 0 exhibited positive clinical manifestations, whereas the majority (92%) of patients classified as Grade 3 demonstrated such clinical signs, indicating a clear trend between increasing Kang grade and clinical severity.

### 3.6. Correlation Between Grading and Clinical Findings

The Kang grading results for 104 hospitalized patients in the test set, evaluated by two resident doctors (Resident 1 and Resident 2), two experienced orthopedic clinicians (Clinician 1 and Clinician 2), and the deep learning model (CNN), are summarized in Table 7. The distribution of each grading category was as follows: Grade 0 accounted for 4–8%, Grade 1 for 29–36%, Grade 2 for 46–51%, and Grade 3 for 12–14%. Compared to the distribution of the entire test set, the proportions of Grade 0 and Grade 1 were relatively lower in the hospitalized patient cohort, while the proportions of Grade 2 and Grade 3 were higher. This pattern is consistent with actual clinical practice, as patients with more severe spinal canal stenosis are more likely to be hospitalized for further management, while those with normal or mildly stenotic cervical spines are typically managed as outpatients.

Furthermore, the proportion of patients presenting with positive clinical signs of cervical spinal canal stenosis increased with higher Kang grades. As expected, none of the patients graded as 0 exhibited positive clinical findings, whereas the vast majority (92%) of those graded as 3 showed positive clinical signs. This trend highlights the clinical relevance of Kang grading in reflecting the severity of cervical spinal canal stenosis.

The association between clinical symptoms and Kang grading was assessed using Spearman’s rank correlation coefficient. The strength of correlation was interpreted as follows: weak correlation for 0.1 < R ≤ 0.3, moderate correlation for 0.3 < R ≤ 0.7, relatively high correlation for 0.7 < R ≤ 0.9, and very high correlation for R > 0.9. A *p* value of < 0.001 was considered statistically significant.

As shown in Table 8, the Spearman correlation analysis for hospitalized patients demonstrated a moderate correlation between the presence of positive clinical signs of cervical spinal stenosis and Kang grades assigned by the two residents (Resident 1 and Resident 2), two orthopedic clinicians (Clinician 1 and Clinician 2), and the deep learning model (CNN). The CNN model achieved a correlation coefficient of R = 0.526, which was comparable to those of the orthopedic clinicians (R = 0.517 and R = 0.503) and higher than those of the residents (R = 0.457 and R = 0.448). These findings indicate that the CNN model’s grading performance was superior to that of the residents and closely aligned with the assessments of experienced orthopedic clinicians.

## 4. Discussion

In this study, we developed and validated a deep-learning-based model for the automated grading of cervical spinal canal stenosis on sagittal T2-weighted MR images using the Kang classification. While the proposed model achieved promising performance, several technical challenges and data-specific issues encountered during model development warrant discussion.

In preliminary experiments, directly inputting raw cervical MR images into deep learning models resulted in a suboptimal classification accuracy of only 60%. We identified several factors that may have contributed to this limitation, including a relatively small dataset size, interference from irrelevant anatomical structures (such as the skull base, paraspinal muscles, and anterior soft tissues), and the presence of multiple spinal segments with different grades of stenosis within a single sagittal image. To address these challenges and enhance model performance, an image preprocessing workflow was implemented. This included converting DICOM images to JPG format, annotating regions of interest (ROIs) focused on the most severely compressed cervical segment, and cropping images to exclude unrelated areas. These steps effectively concentrated critical imaging features and minimized noise, which significantly improved the model’s grading performance in subsequent training.

The diagnosis of DCM requires consistency between clinical symptoms and imaging findings [2]. Imaging evidence of spinal cord compression due to cervical canal stenosis is a prerequisite for diagnosing DCM. However, due to discrepancies between imaging findings of spinal cord compression and clinical myelopathic symptoms [5], there is a need for more detailed grading and even quantitative analysis of cervical canal stenosis.

Cervical MRI is a crucial tool for evaluating cervical canal stenosis, yet there is no widely accepted MR grading system. Several studies have proposed MRI-based grading for cervical canal stenosis. Muhle et al. [26] included subarachnoid space occlusion in their grading system, while Larsson et al. [27] quantified the degree of subarachnoid space obstruction into mild, moderate, and severe categories. However, neither grading system incorporated T2 signal changes.

Studies have shown that high T2 signal intensity is a strong predictor of late-stage progression in PSCCC [6]. Kang et al. [9] incorporated T2 high signal intensity into cervical canal stenosis grading, making it more clinically relevant. Park [10] further modified Kang’s grading by classifying subarachnoid space occlusion <50% as grade 0 and studied the correlation between the modified grading system and the severity of neurological impairment. Since the presence of radicular symptoms is also an effective early predictor of PSCCC progression [5], the Kang grading system has greater clinical significance, particularly for PSCCC patients without myelopathic symptoms undergoing conservative treatment, as it aids in assessing the risk of progression to DCM. However, LEE et al. [11] reported a relatively low interobserver agreement (*κ* = 0.691) among clinicians using the Kang grading system, which is consistent with our findings.

The application of deep learning in cervical canal stenosis is still limited. Zamir et al. [28] developed a deep learning model called ResNet50 for the binary classification of cervical spinal cord compression on axial T2-weighted MRI, achieving an overall AUC of 0.94. However, the binary classification approach restricts the model’s clinical utility. Additionally, their study only included images from patients with a confirmed diagnosis of cervical myelopathy, excluding normal spinal canals and cases with mild stenosis, thereby reducing the model’s applicability, particularly for mildly symptomatic outpatients.

Most studies on the Kang grading system have excluded patients with foraminal stenosis. Our study extends the Kang grading system to include all patients, evaluating its applicability, and demonstrates excellent interobserver agreement.

The interobserver agreement for Kang grading among junior physicians in our study (*κ* = 0.684) was similar to that reported by LEE (*κ* = 0.691) and was lower than that of senior physicians (*κ* = 0.926) or radiologists (Lee, *κ* = 0.912). This finding suggests that extensive clinical experience is required to accurately apply Kang grading, which limits its widespread use in clinical practice.

In this study, we evaluated the consistency between Kang grading results generated by a deep learning classification model and those assigned by clinicians, as well as their correlation with clinical manifestations of cervical spinal stenosis. The CNN model achieved a Spearman correlation coefficient of 0.526 with clinical symptoms, which was comparable to the values observed for experienced orthopedic clinicians (0.517 and 0.503) and higher than those of residents (0.457 and 0.448). These findings suggest that the deep learning model not only approximates the diagnostic performance of seasoned clinicians but also exceeds that of less experienced physicians in evaluating the clinical severity of cervical spinal stenosis.

The use of Spearman’s rank correlation allowed for a non-parametric assessment of the monotonic relationship between imaging-based grading and clinical presentation. The moderate correlation observed in this study reflects the inherent complexity and multifactorial nature of cervical spinal stenosis, where imaging findings do not always fully predict symptom severity. Nonetheless, the alignment of model-derived grading with clinical findings supports its potential as a reliable adjunctive tool in clinical decision-making.

Importantly, the CNN model demonstrated consistent performance across different severity levels, maintaining a clinically meaningful correlation with symptomatology. This highlights its potential for assisting clinicians in initial triage, follow-up evaluation, and even educational training. As deep learning technologies continue to evolve, integrating them into routine radiological workflows may improve diagnostic efficiency and reduce interobserver variability, particularly in high-volume or resource-limited settings.

### Limitations and Prospects

First, the images underwent preprocessing, requiring manual selection of the mid-sagittal plane and region of interest (ROI) extraction. Only the most severely stenotic segment was graded. Future research could incorporate an object detection model for automated ROI localization and the segmentation of cervical canal stenosis at each level. Additionally, this study is retrospective, and neurological symptoms were not quantitatively assessed but simply categorized as positive or negative. Finally, only sagittal T2-weighted images were analyzed, without incorporating axial views for a more comprehensive evaluation of cervical canal stenosis.

Future studies should consider incorporating multimodal data, such as patient-reported symptoms and CT imaging, as well as extending the approach to longitudinal monitoring and integration into clinical workflows through decision support systems. Novel architectures incorporating multimodal or multi-channel inputs by the application of three-dimensional convolutional neural networks (3D CNNs) could enable more comprehensive spatial feature extraction, potentially enhancing the accuracy and robustness of cervical spinal canal stenosis grading models [29,30]. Furthermore, establishing an external validation cohort from independent healthcare institutions would help enhance the generalizability and clinical applicability of the model.

## 5. Conclusions

This study developed a deep-learning-based intelligent Kang grading system for cervical spinal canal stenosis, which is supported by radiculopathy symptoms. The model was trained on a large dataset with relatively low inter-reader consistency from junior physicians and demonstrated improved grading performance on the test set, which can be beneficial to play a role in assisting clinicians in clinical practical work and also provides a reliable and convenient grading tool for further high-quality clinical studies related to the grading of cervical spinal stenosis in the future. It is recommended that the outputs of each individual model be carefully interpreted to effectively evaluate explainability. Moreover, future studies should address the principal limitations of this study to enhance the model’s generalizability and its potential for clinical translation.

## Figures and Tables

**Figure 1 bioengineering-12-00604-f001:**
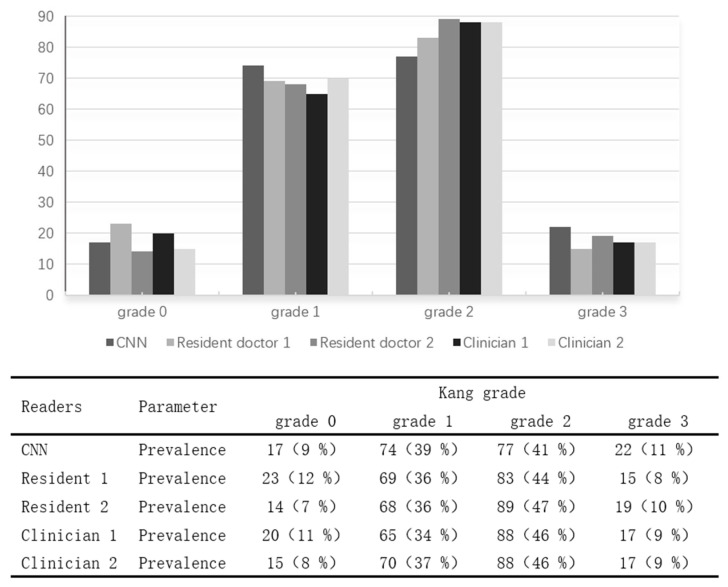
The distribution of Kang grading for the test set of readers.

**Table 1 bioengineering-12-00604-t001:** Confusion matrix Between reader1 and reader2 in the training/validation datasets.

Kang Grade	Resident 1				Kappa Value	*p* Value
	Grade 0	Grade 1	Grade 2	Grade 3		
Resident 2						
Grade 0	127	15	0	0	0.672	<0.001
Grade 1	42	219	30	0		
Grade 2	0	62	202	12		
Grade 3	0	0	12	43		

**Table 2 bioengineering-12-00604-t002:** Average accuracy of five-fold cross-validation for training/validation sets of basic classification models.

Model Architecture	Validation Accuracy (Five-Fold) Mean/SD
GoogLeNet	64.01/4.5%
ResNet50	69.37/3.5%
ResNeXt50	65.04/6.3%
DesNet121	67.55/4.7%
EfficientNet_B0	77.89/2.6%

**Table 3 bioengineering-12-00604-t003:** Average accuracy of five-fold cross-validation for training/validation sets of EfficientNet classification models.

Model Architecture	Validation Accuracy (Five-Fold) Mean/SD
EfficientNet_B0	77.89/2.6%
EfficientNet_B1	78.02/3.5%
EfficientNet_B2	78.79/1.6%
EfficientNet_B3	78.80/2.3%
EfficientNet_B4	78.68/3.8%
EfficientNet_B5	79.45/2.1%
EfficientNet_B6	78.15/1.7%
EfficientNet_B7	76.32/1.2%

**Table 4 bioengineering-12-00604-t004:** Grading performance of EfficientNet_B5 in the test set (clinician1 as a relative standard).

Predicted Grading	Precision (%)	F1 Score (%)	Accuracy (%)	Sensitivity (%)	Specificity (%)
Grade 0	100	91.9	98.4	85.0	100
Grade 1	85.1	90.6	92.6	96.9	91.2
Grade 2	97.4	90.9	92.1	85.2	98.0
Grade 3	72.7	82.1	96.3	94.1	96.5
Weighted	91.2	90.1	93.3	90.0	96.1

**Table 5 bioengineering-12-00604-t005:** Grading performance of EfficientNet_B5 in the test set (clinician2 as a relative standard).

Predicted Grading	Precision (%)	F1 Score (%)	Accuracy (%)	Sensitivity (%)	Specificity (%)
Grade 0	82.4	87.5	97.9	93.3	98.3
Grade 1	86.5	88.8	91.6	91.4	91.6
Grade 2	96.1	89.7	89.7	84.1	97.1
Grade 3	72.7	82.1	96.3	94.1	96.5
Weighted	89.4	88.4	92.3	88.4	95.1

**Table 6 bioengineering-12-00604-t006:** Kappa consistency analysis of Kang grading for the test set of readers.

Readers	Kappa Value	Standard Error
Clinician 1 vs. Clinician 2	0.926	0.024
Resident 1 vs. Resident 2	0.684	0.045
CNN vs. Clinician 1	0.848	0.033
CNN vs. Clinician 2	0.822	0.036
Resident 1 vs. Clinician 1	0.758	0.041
Resident 2 vs. Clinician 1	0.698	0.045
Resident 1 vs. Clinician 2	0.732	0.043
Resident 2 vs. Clinician 2	0.702	0.045
CNN vs. Resident 1	0.658	0.046
CNN vs. Resident 2	0.613	0.049

**Table 7 bioengineering-12-00604-t007:** Distribution of Kang grade and positive clinical manifestation for inpatients in test set.

Readers	Parameter	Kang Grade			
		Grade 0	Grade 1	Grade 2	Grade 3
CNN	Prevalence	4 (4%)	37 (36%)	48 (46%)	15 (14%)
	Positive clinical manifestations	0 (0%)	7 (19%)	26 (54%)	14 (93%)
Resident1	Prevalence	8 (8%)	30 (29%)	53 (51%)	13 (12%)
	Positive clinical manifestations	0 (0%)	8 (26%)	27 (51%)	12 (92%)
Resident2	Prevalence	6 (6%)	30 (29%)	53 (51%)	15 (14%)
	Positive clinical manifestations	0 (0%)	8 (27%)	25 (47%)	14 (93%)
Clinician1	Prevalence	7 (7%)	34 (33%)	50 (48%)	13 (12%)
	Positive clinical manifestations	0 (0%)	7 (21%)	28 (56%)	12 (92%)
Clinician2	Prevalence	5 (5%)	35 (34%)	51 (49%)	13 (12%)
	Positive clinical manifestations	0 (0%)	7 (20%)	28 (55%)	12 (92%)

**Table 8 bioengineering-12-00604-t008:** Correlation analysis between Kang grade and positive clinical manifestations of cervical spinal stenosis.

Readers	Correlation Coefficients	*p* Value
Clinician 1	0.517	<0.001
Clinician 2	0.503	<0.001
CNN	0.526	<0.001
Resident 1	0.457	<0.001
Resident 2	0.448	<0.001

## Data Availability

The original contributions presented in this study are included in the article. Further inquiries can be directed to the corresponding author(s).

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
