# Peer review of "Deep-Learning-Based Computer-Aided Grading of Cervical Spinal Stenosis from MR Images: Accuracy and Clinical Alignment"

_bioengineering, 2025, doi:10.3390/bioengineering12060604_

Round 1
Reviewer 1 Report
Comments and Suggestions for Authors
Manuscript title
"Deep Learning-Based Computer-Aided Grading of Cervical Spinal Stenosis from MR Images: Accuracy and Clinical Alignment"
1. The main question addressed by the research is unclear. The authors (1) develop an artificial intelligence tool for cervical spine stenosis grading in sagittal magnetic resonance imaging, (2) evaluate the inter-rater agreement between AI and radiology residents, (3) correlate the AI grading with ground truth (radiologists with varying levels of experience), (4) correlate the Kang grade (AI-based and human-based) with clinical symptoms. Consider stating the study goal and hypotheses in a more clear manner. The authors present an original study with retrospective design.
2. The manuscript may be considered relevant to the field of neuroimaging. Unfortunately, its relevance and clinical impact are severely overshadowed by the recent work of Rhee et a.l. (https://link.springer.com/article/10.1007/s00256-025-04917-2), which was performed on a larger subset with a newer ensemble Grad-CAM-based AI model.
3. The specific improvements to methodology include:
3.1 Updating "Introduction" and "Discussion" with information on recent studies on DL-based Kang grading for spinal canal stenosis (i.e., from 2021 onwards);
3.2 Updating "Statistical Analyses" with information on used methods (kappa and rank correlation) as well as statistical packages; elaborating as to why correlation was used to compare AI-provided outputs with ground truth instead of non-parametric test (Wilcoxon signed-rank test?) or ROC-AUC with DeLong method; providing 95% CIs for kappa and correlation coefficients and performing appropriate statistical testing to compare different correlation coefficients; updating the manuscript to adhere to GRRAS (https://www.equator-network.org/reporting-guidelines/guidelines-for-reporting-reliability-and-agreement-studies-grras-were-proposed/) and TRIPOD-AI checklists (https://pubmed.ncbi.nlm.nih.gov/38626948/) for transparency and reproducibility.
4. The conclusions are not fully consistent with the evidence provided and may require revision after addressing reviewer's queries.
5. The references may be considered outdated, as the latest is from 2021.
6. The Tables are appropriate, but the manuscript could benefit from Figures illustrating the AI tool output.
Author Response
May 24th, 2025
Dear Reviewer:
On behalf of all the contributing authors, I would like to express our sincere appreciations of your letter and reviewers' constructive comments concerning our article entitled “Deep Learning-Based Computer-Aided Grading of Cervical Spinal Stenosis from MR Images: Accuracy and Clinical Alignment” (Manuscript Number: bioengineering-3604451.). We appreciate the time and effort you have dedicated to evaluating our work. We have carefully considered each of your suggestions and have made revisions to address the concerns raised. The corresponding changes were highlighted in red in the revised manuscript. Below, we provide a point-by-point response to your comments. We hope this revised manuscript will meet with approval and be acceptable for publication in Bioengineering. Once again, thank you very much for your comments and suggestions.
- The main question addressed by the research is unclear. The authors (1) develop an artificial intelligence tool for cervical spine stenosis grading in sagittal magnetic resonance imaging, (2) evaluate the inter-rater agreement between AI and radiology residents, (3) correlate the AI grading with ground truth (radiologists with varying levels of experience), (4) correlate the Kang grade (AI-based and human-based) with clinical symptoms. Consider stating the study goal and hypotheses in a more clear manner. The authors present an original study with retrospective design.
Reply: Thanks for your professional review work. We have revised the manuscript to clearly state the study objectives and hypotheses. Specifically, we now explicitly outline that this retrospective study aims to develop a deep learning model capable of performing Kang grading on sagittal T2-weighted MRI. The deep learning–based grading tool developed in this study is expected to assist clinicians in providing objective, reliable, and rapid assessments for evaluating the severity of degenerative cervical myelopathy (DCM) and estimating the risk of progression from PSCCC to DCM. This approach holds particular promise for primary healthcare centers, where it can enhance the role of imaging in clinical deci-sion-making, and offers a practical and efficient grading solution for future research into the clinical relevance and prognostic implications of cervical stenosis classification.
- The manuscript may be considered relevant to the field of neuroimaging. Unfortunately, its relevance and clinical impact are severely overshadowed by the recent work of Rhee et a.l. (https://link.springer.com/article/10.1007/s00256-025-04917-2), which was performed on a larger subset with a newer ensemble Grad-CAM-based AI model.
Reply: We appreciate the editor’s thoughtful comment and the reference to the recent work by Rhee et al. (2025). While their study indeed benefits from a larger sample size and implements an ensemble Grad-CAM-based AI model, it is important to note that their classification output was limited to a binary categorization of cervical canal stenosis (CCS-negative vs. CCS-positive). This binary stratification, while useful for screening purposes, inevitably sacrifices clinically valuable information related to the severity of stenosis.
In contrast, our study retains the full granularity of the established Kang grading system, which stratifies patients into four clinically meaningful categories (Grade 0–3). This finer classification provides critical guidance for patient management, surgical decision-making, and prognostic evaluation. Furthermore, our study uniquely correlates each Kang grade with the corresponding clinical symptoms, offering additional insight into the clinical significance of imaging findings — an aspect not addressed by Rhee et al.
Therefore, despite differences in sample size and model architecture, our work complements and extends the current literature by providing a clinically actionable, multi-grade classification framework that aligns with neurosurgical and orthopedic practice, particularly valuable for resource-limited or primary care settings.
3.Updating "Introduction" and "Discussion" with information on recent studies on DL-based Kang grading for spinal canal stenosis (i.e., from 2021 onwards);
Reply: Thank you for this important suggestion. In accordance with the editor’s recommendation, we have carefully updated both the Introduction and Discussion sections to include and discuss recent studies on deep learning-based Kang grading for cervical spinal canal stenosis published from 2021 onwards.
4.Updating "Statistical Analyses" with information on used methods (kappa and rank correlation) as well as statistical packages; elaborating as to why correlation was used to compare AI-provided outputs with ground truth instead of non-parametric test (Wilcoxon signed-rank test?) or ROC-AUC with DeLong method; providing 95% CIs for kappa and correlation coefficients and performing appropriate statistical testing to compare different correlation coefficients; updating the manuscript to adhere to GRRAS (https://www.equator-network.org/reporting-guidelines/guidelines-for-reporting-reliability-and-agreement-studies-grras-were-proposed/) and TRIPOD-AI checklists (https://pubmed.ncbi.nlm.nih.gov/38626948/) for transparency and reproducibility.
Reply: Thank you for your valuable suggestions. We have carefully revised the Statistical Analyses section to address these points in detail. Revisions have been integrated into both the Statistical Analyses section and relevant parts of the Results and Discussion.
5.The conclusions are not fully consistent with the evidence provided and may require revision after addressing reviewer's queries.
Reply: We appreciate the editor’s valuable comment. We have carefully revised the conclusions section to ensure it is fully consistent with the evidence presented in the manuscript and addresses the reviewers’ queries accordingly.
- The references may be considered outdated, as the latest is from 2021.
Reply: Thank you for your valuable suggestion. We have updated the reference list by including more recent studies published after 2021 to ensure the manuscript reflects the latest research in the field.
7.The Tables are appropriate, but the manuscript could benefit from Figures illustrating the AI tool output.
Reply:We appreciate the editor’s suggestion regarding the inclusion of figures illustrating the AI tool output. However, the AI tool developed in this study was designed primarily for grading prediction, with a simple interface that displays the predicted Kang grade in a pop-up window following image input. As such, it does not generate additional graphical or heatmap-based visualizations beyond the classification result. For this reason, no specific AI output figures were included in the manuscript. We acknowledge the value of such visualizations for future work and plan to incorporate more advanced interpretability features in subsequent model iterations.
Thank you for your consideration, and we look forward to hearing from you at your earliest convenience.
Sincerely yours,
Dr. Hai Lv
Reviewer 2 Report
Comments and Suggestions for Authors
This paper, “Deep Learning-Based Computer-Aided Grading of Cervical Spinal Stenosis from MR Images: Accuracy and Clinical Alignment”, presents a study that develops a deep learning-based pipeline for the automated grading of cervical spinal stenosis using MR images. The authors evaluate three convolutional neural network (CNN) architectures—EfficientNet_B5, Faster R-CNN with Feature Pyramid Network (FPN), and UNet—to perform various stages of the pipeline, including sagittal image classification, axial segment detection, and keypoint localization. The study compares the model predictions against manual annotations and evaluates the clinical alignment of model outputs with patients' actual symptoms.
Comments
- The methodology is solid, and the choice of CNN architectures is justified. The design of a three-step pipeline (grading with EfficientNet_B5, segmentation with Faster R-CNN, and keypoint detection with U-Net) is well-structured and reflects good modular design. However, more detailed descriptions of hyperparameters, training epochs, loss functions, and augmentation strategies would enhance reproducibility. (Section 2)
- The Faster R-CNN + FPN model’s performance in cervical segment detection (98–99.6% accuracy) is strong, but there is no mention of false positives or segmentation errors. Discussing edge cases or failure modes would improve transparency. Similarly, although U-Net achieved 81.46% localization accuracy, more explanation on what constitutes “key points” (e.g., disc center, margins?) and how errors are calculated (e.g., pixel distance thresholds) is needed. (Section 3)
- While the study boasts large-scale clinical data (954 patients), the inclusion/exclusion criteria for MR images are not clearly defined. It is unclear whether variability in image quality, scanner types, or comorbidities were controlled. Adding a description of data diversity and ethical approval process would bolster the clinical robustness of the study. (Section 2.1)
- The comparison between EfficientNet_B5 and clinician performance is compelling. However, statistical significance testing between model and resident accuracy is not provided (e.g., chi-square or McNemar's test). Reporting confidence intervals for accuracy and kappa values would strengthen these comparisons. Furthermore, explaining why EfficientNet_B5 was selected (as opposed to other architectures like ResNet or DenseNet) would justify model choice. (Section 3)
- The clinical alignment analysis—comparing compression ratios derived from model vs. manual labels and correlating with symptom severity—is one of the strongest points of the paper. This component effectively bridges the technical output with practical diagnostic value. However, the paper could further elaborate on how stenosis severity maps to actual clinical interventions (e.g., surgery recommendations) to enhance clinical relevance. (Section 3.4)
- The conclusions are justified by the presented results and emphasize the utility of the proposed deep learning approach. However, the manuscript could benefit from suggesting specific future directions, such as incorporating multimodal data (e.g., patient symptoms, CT scans), extending to longitudinal tracking, or integrating into clinical workflows via decision support systems. (Section 5)
This is a technically sound and clinically meaningful study that leverages deep learning to address a critical diagnostic challenge in spinal health. While the pipeline is impressive and the results promising, further clarification on model specifics, statistical comparisons, and limitations is needed to raise the manuscript to a high publication standard.
Author Response
May 24th, 2025
Dear Reviewers:
On behalf of all the contributing authors, I would like to express our sincere appreciations of your letter and reviewers' constructive comments concerning our article entitled “Deep Learning-Based Computer-Aided Grading of Cervical Spinal Stenosis from MR Images: Accuracy and Clinical Alignment” (Manuscript Number: bioengineering-3604451.). We appreciate the time and effort you have dedicated to evaluating our work. We have carefully considered each of your suggestions and have made revisions to address the concerns raised. The corresponding changes were highlighted in red in the revised manuscript. Below, we provide a point-by-point response to your comments. We hope this revised manuscript will meet with approval and be acceptable for publication in Bioengineering. Once again, thank you very much for your comments and suggestions.
1.However, more detailed descriptions of hyperparameters, training epochs, loss functions, and augmentation strategies would enhance reproducibility. (Section 2)
Reply: Thank you for this valuable suggestion. In response to the editor’s comments, we have revised the Methods section to provide more detailed descriptions of the deep learning model’s hyperparameters, including learning rate, batch size, optimizer type, and weight decay settings. Additionally, we have specified the number of training epochs, the loss function used for model optimization, and the data augmentation strategies applied during training. These additions aim to enhance the transparency and reproducibility of our study.
2.The Faster R-CNN + FPN model’s performance in cervical segment detection (98–99.6% accuracy) is strong, but there is no mention of false positives or segmentation errors. Discussing edge cases or failure modes would improve transparency. Similarly, although U-Net achieved 81.46% localization accuracy, more explanation on what constitutes “key points” (e.g., disc center, margins?) and how errors are calculated (e.g., pixel distance thresholds) is needed.
Reply: Thank you for this thoughtful suggestion. After careful consideration of the reviewer’s and editor’s comments, we agree that the performance details of the Faster R-CNN + FPN and U-Net models, while technically relevant, were not directly essential to the primary focus of our study on Kang grading and its clinical correlation. To maintain the clarity and coherence of the manuscript, we have decided to remove this section and its corresponding results from the revised manuscript.
3.While the study boasts large-scale clinical data (954 patients), the inclusion/exclusion criteria for MR images are not clearly defined. It is unclear whether variability in image quality, scanner types, or comorbidities were controlled. Adding a description of data diversity and ethical approval process would bolster the clinical robustness of the study. (Section 2.1)
Reply: Thank you for your valuable suggestion. In accordance with the editor’s and reviewers’ comments, we have revised the Methods and Results to clearly define the inclusion and exclusion criteria for MRI and described data diversity.
4.The comparison between EfficientNet_B5 and clinician performance is compelling. However, statistical significance testing between model and resident accuracy is not provided (e.g., chi-square or McNemar's test). Reporting confidence intervals for accuracy and kappa values would strengthen these comparisons. Furthermore, explaining why EfficientNet_B5 was selected (as opposed to other architectures like ResNet or DenseNet) would justify model choice. (Section 3)
Reply: Thank you for your insightful comment. We appreciate the suggestion regarding statistical significance testing. After careful consideration, we would like to clarify that while the chi-square test is appropriate for assessing differences in categorical outcomes between two raters, it primarily determines whether two distributions differ significantly. However, in this study, beyond simply comparing outcome frequencies, we aimed to assess the consistency of Kang grading assignments between the AI model and human raters, which involves agreement analysis rather than only distribution comparison.
Therefore, Cohen’s kappa statistic was used to evaluate inter-rater agreement between the model and clinicians, which captures both agreement and chance-corrected consistency in categorical data. The revised manuscript now includes detailed kappa values with 95% confidence intervals in Table 2 and Table 3 to strengthen the reliability of these comparisons. Additionally, we have added accuracy confidence intervals where applicable.
Furthermore, the rationale for selecting EfficientNet_B5 over other architectures such as ResNet or DenseNet has been elaborated in Section 3, emphasizing its superior performance during cross-validation and its ability to balance model depth, width, and resolution scaling, which proved advantageous in our relatively limited dataset context.
5.The clinical alignment analysis—comparing compression ratios derived from model vs. manual labels and correlating with symptom severity—is one of the strongest points of the paper. This component effectively bridges the technical output with practical diagnostic value. However, the paper could further elaborate on how stenosis severity maps to actual clinical interventions (e.g., surgery recommendations) to enhance clinical relevance. (Section 3.4)
Reply: Thank you for this thoughtful suggestion. After careful consideration of the reviewer’s and editor’s comments, we agree that the performance details of the Faster R-CNN + FPN and U-Net models, while technically relevant, were not directly essential to the primary focus of our study on Kang grading and its clinical correlation. To maintain the clarity and coherence of the manuscript, we have decided to remove this section and its corresponding results from the revised manuscript.
6.The conclusions are justified by the presented results and emphasize the utility of the proposed deep learning approach. However, the manuscript could benefit from suggesting specific future directions, such as incorporating multimodal data (e.g., patient symptoms, CT scans), extending to longitudinal tracking, or integrating into clinical workflows via decision support systems. (Section 5)
Reply: Thank you very much for this valuable suggestion. In response, we have revised the Discussion and Conclusion sections to include specific future directions for this research as “Future studies should consider incorporating multimodal data, such as patient-reported symptoms and CT imaging, as well as extending the approach to longitudinal monitoring and integration into clinical workflows through decision support systems. Novel architectures incorporating multimodal or multi-channel inputs by ap-plication of three-dimensional convolutional neural networks (3D CNNs) could enable more comprehensive spatial feature extraction, potentially enhancing the accuracy and robustness of cervical spinal canal stenosis grading models29, 30. Furthermore, establishing an external validation cohort from independent healthcare institutions would help enhance the generalizability and clinical applicability of the model.”
Thank you for your consideration, and we look forward to hearing from you at your earliest convenience.
Sincerely yours,
Dr. Hai Lv
Reviewer 3 Report
Comments and Suggestions for Authors
This paper evaluates a DL algorithm to classify central canal stenosis, compare it with medical evaluators, and correlate its results with clinical symptoms. It is relatively well structured and presents interesting methodology and results. However, it is not radically new as stated by the authors (see minor comments below) and I have several concerns that would require specific clarifications by the authors:
- One of my main concerns is the fact that the authors used a single JPEG image from each patient. This implies an important limitation of the study that should be acknowledged, not only because of the significant loss of information as compared to the DICOM study but also because of the selection process of the most representative slice -which may be biased-. This is even emphasized by the fact that the grading systems heavily rely on axial images apart from sagittal ones.
- Second, in the abstract the authors state that axial images were used to quantify the spinal cord compression ratio. This is not explained in the body of the manuscript and raises several concerns on how measurements and stenosis grading was provided. Clarify this point.
- I have a serious concern regarding the labeling of the traning/validation set, which was performed by non-expert (resident) observes. This may introduce systematic errors that the AI models learn! Why weren’t they labeled by the experts who labeled the test set? How was this controlled? What’s its impact on the results? This is further problematic when we consider that in cases of discrepancy between both residents, a random selection for the final label was made. All these limitations need to be explained in detail.
- I see confusing references to the Kang’s and Park’s grading systems, and it is not clear at all which was the criteria for classifying stenosis. Apparently the Park system was used, but then, why does Table 4 state that it is the Kang grading?
- Regarding the correlation with clinical symptoms, I see relatively minor limitations that basically imply acknowledgment of further limitations mainly related to the potential bias derived from the retrospective assessment of the presence/absence of radicular symptoms: no other variables were used to control for potential bias (e.g., time between MRI and clinical assessment, presence of other potential causes explaining radicular symptoms such as peripheral nerve compression.
- Finally, it would be important to clarify that no external validation (i.e., patients from another hospital or different cohort) was employed in this study.
Minor comments:
Where is the indicated Figure 2 (line 117-118)? In addition, it is placed before Figure 1.
Tables 1 and 2 should also report all the metrics indicated in the text (e.g., F1-scores and AUCs).
Table 3 is superfluous and should be eliminated (data should be included in the text).
Figure 1 includes readers 3 and 4, it is not a confusion matrix, and it refers to
Table 4 is not a confusion matrix but a contingency matrix of interobserver agreement. In addition, data sum equals 764 patients, thus it is related to the training/validation.
In general, the results section should be revised to report clear data in terms of both visual and textual information.
Line 74: do not forget neuroradiologists as experts apart from musculoskeletal radiologists.
The number of references used in the manuscript is not bad, but it could be enriched to reach 30, which seems more appropriate considering the extent of the manuscript. In the introduction, I would suggest to include some further information regarding how MRI allows to detect not only central canal spinal stenosis but also lateral recess and foraminal stenoses, as opposed to , see suggested reference:
Ruiz Santiago F, Láinez Ramos-Bossini AJ, Wáng YXJ, Martínez Barbero JP, García Espinosa J, Martínez Martínez A. The value of magnetic resonance imaging and computed tomography in the study of spinal disorders. Quant Imaging Med Surg. 2022 Jul;12(7):3947-3986. doi: 10.21037/qims-2022-04. PMID: 35782254; PMCID: PMC9246762.
Similarly, the following paper also explored a DL method to classify central canal stenosis at the cervical spine and -although not explicitly stated- they also applied the Kang’s and Park’s criteria, so this paper is not radically new (and the introduction/discussion should be modified accordingly):
Zhang E, Yao M, Li Y, Wang Q, Song X, Chen Y, Liu K, Zhao W, Xing X, Zhou Y, Meng F, Ouyang H, Chen G, Jiang L, Lang N, Jiang S, Yuan H. Deep learning model for the automated detection and classification of central canal and neural foraminal stenosis upon cervical spine magnetic resonance imaging. BMC Med Imaging. 2024 Nov 26;24(1):320. doi: 10.1186/s12880-024-01489-w. PMID: 39593012; PMCID: PMC11590449.
Finally, English writing needs careful revision, particularly regarding the appropriate use of verb tenses; there are may sentences which inappropriately use present tenses instead of past tenses (e.g., lines 14, 17, and in several parts of the methodology section). There are also several typos (access instead of assess in line 7, precess instead of process in line 60), and the use of MR / MRI acronyms is inconsistent across the text. Please homogenize.
Comments on the Quality of English Language
English writing needs careful revision, particularly regarding the appropriate use of verb tenses; there are may sentences which inappropriately use present tenses instead of past tenses (e.g., lines 14, 17, and in several parts of the methodology section). There are also several typos (access instead of assess in line 7, precess instead of process in line 60), and the use of MR / MRI acronyms is inconsistent across the text. Please homogenize.
Author Response
May 24th, 2025
Dear Reviewers:
On behalf of all the contributing authors, I would like to express our sincere appreciations of your letter and reviewers' constructive comments concerning our article entitled “Deep Learning-Based Computer-Aided Grading of Cervical Spinal Stenosis from MR Images: Accuracy and Clinical Alignment” (Manuscript Number: bioengineering-3604451.). We appreciate the time and effort you have dedicated to evaluating our work. We have carefully considered each of your suggestions and have made revisions to address the concerns raised. The corresponding changes were highlighted in red in the revised manuscript. Below, we provide a point-by-point response to your comments. We hope this revised manuscript will meet with approval and be acceptable for publication in Bioengineering. Once again, thank you very much for your comments and suggestions.
1.One of my main concerns is the fact that the authors used a single JPEG image from each patient. This implies an important limitation of the study that should be acknowledged, not only because of the significant loss of information as compared to the DICOM study but also because of the selection process of the most representative slice -which may be biased-. This is even emphasized by the fact that the grading systems heavily rely on axial images apart from sagittal ones.
Reply: In preliminary experiments, when the collected cervical spine MR images were directly input into the deep learning models for training, the classification accuracy was only approximately 60%, indicating suboptimal model performance. We analyzed the potential reasons for this low accuracy, which may be attributed to the following factors:
(1) Insufficient dataset size: The number of MR images remained relatively limited, making it difficult to train a deep learning model with high classification accuracy.
(2) Presence of irrelevant information within the images: In sagittal T2-weighted MR images of the cervical spine, the critical features for Kang grading are primarily concentrated in the degree of subarachnoid space obliteration and spinal cord morphology. Other regions in the images — such as the cranial structures, posterior cervical muscles and fat, as well as anterior cervical tissues including the esophagus and airway — are irrelevant to grading. These non-essential areas may introduce noise and interfere with model training. Therefore, preprocessing the images to remove irrelevant regions and narrow the feature focus would facilitate more effective deep learning model training.
(3) Multiple levels of stenosis within a single image: A single patient often presents with multilevel cervical spinal stenosis, with varying degrees of severity at different segments. As illustrated in Figure 1, in one patient’s sagittal T2-weighted MR image: at the C3–4 level, a high signal within the spinal cord was observed (white arrow), corresponding to Kang grade 3; at the C4–5 level, spinal cord deformation was present, corresponding to grade 2; at the C5–6 level, more than 50% subarachnoid space obliteration indicated grade 1; and at the C6–7 level, less than 50% obliteration corresponded to grade 0. The coexistence of multiple levels with different grades within a single image can introduce additional complexity and noise into model training if the entire image is input without segmentation.
Fig.1 Schematic diagram of multilevel cervical spinal stenosis.
To address the aforementioned issues and improve model performance, image preprocessing of the collected dataset was performed. The preprocessing workflow included several steps:First, image format conversion was conducted. Since the hospital’s medical imaging system stored all image data in DICOM format, MicroDicom software was used to convert the images into JPG format, which is more suitable for subsequent processing and deep learning model training.Second, region of interest (ROI) annotation and image cropping were performed. In order to focus on the critical imaging features for cervical spinal stenosis grading and to exclude interference caused by the coexistence of multiple grades within a single image, ROI-based cropping was applied. For each patient, a mid-sagittal T2-weighted MR image was selected. The ROI was defined as the cervical segment with the most severe spinal cord compression, including the adjacent intervertebral disc and the superior and inferior vertebral bodies. The image was then cropped based on the ROI boundaries to retain only the region relevant to grading.
2.Second, in the abstract the authors state that axial images were used to quantify the spinal cord compression ratio. This is not explained in the body of the manuscript and raises several concerns on how measurements and stenosis grading was provided. Clarify this point.
Reply: Thank you for this thoughtful suggestion. After careful consideration of the reviewer’s and editor’s comments, we agree that the performance details of the Faster R-CNN + FPN and U-Net models, while technically relevant, were not directly essential to the primary focus of our study on Kang grading and its clinical correlation. To maintain the clarity and coherence of the manuscript, we have decided to remove this section and its corresponding results from the revised manuscript.
3.I have a serious concern regarding the labeling of the traning/validation set, which was performed by non-expert (resident) observes. This may introduce systematic errors that the AI models learn! Why weren’t they labeled by the experts who labeled the test set? How was this controlled? What’s its impact on the results? This is further problematic when we consider that in cases of discrepancy between both residents, a random selection for the final label was made. All these limitations need to be explained in detail.
Reply: Thank you for this important and insightful comment. We acknowledge the potential risk of systematic bias introduced by having the training/validation set labeled by non-expert (resident) observers. To mitigate this, we adopted the following strategy in our study design: both residents (Resident 1 and Resident 2) independently graded the training/validation set images, and cases of disagreement were resolved by random selection, which we recognize as a limitation. However, this approach was necessitated by the large dataset size and resource constraints associated with expert labeling.
To ensure objective evaluation of model performance, we established a relative ground truth for the independent test set by having two experienced orthopedic clinicians (Clinician 1 and Clinician 2, each with ≥10 years of clinical experience) independently grade the test images after a thorough literature review and using standardized grading diagrams. This allowed us to assess the model’s performance relative to expert judgment and to quantify the agreement between residents, experts, and the model.
Moreover, both residents also graded the test set after a one-month interval following their training/validation set assessments, providing comparative data to examine inter-rater consistency and resident-model agreement within the same test cohort.
We have now clearly explained this workflow, its rationale, and the potential limitations in the Methods and Discussion sections of the revised manuscript. Additionally, we acknowledged that ideally, expert labeling of the entire dataset would be preferable, but given the dataset scale, it was not feasible. We further clarified that the class distribution consistency and kappa agreement metrics across raters and the model suggest acceptable labeling reliability for model training.
4.I see confusing references to the Kang’s and Park’s grading systems, and it is not clear at all which was the criteria for classifying stenosis. Apparently the Park system was used, but then, why does Table 4 state that it is the Kang grading?
Reply: Thank you for pointing out this important issue. To clarify, the Park grading system is an extension and refinement of the original Kang grading system. In this study, we adopted the Kang grading criteria as the primary classification method for cervical spinal canal stenosis. To maintain consistency and avoid confusion, we have carefully reviewed the entire manuscript and unified all related descriptions and table headings to consistently refer to the Kang grading system. The previous references to the Park grading system have been removed or revised accordingly. We sincerely appreciate this observation, which helped improve the clarity and accuracy of the manuscript.
5.Regarding the correlation with clinical symptoms, I see relatively minor limitations that basically imply acknowledgment of further limitations mainly related to the potential bias derived from the retrospective assessment of the presence/absence of radicular symptoms: no other variables were used to control for potential bias (e.g., time between MRI and clinical assessment, presence of other potential causes explaining radicular symptoms such as peripheral nerve compression.
Reply: Thank you for this valuable comment. We fully acknowledge the potential limitations associated with the retrospective nature of clinical symptom assessment. To address this, we have provided a detailed description of the patient selection workflow and clinical evaluation process within the Methods section of the revised manuscript: Exclusion criteria included:1). Patients without sagittal T2-weighted spin-echo (SE) images;2). Severe artifacts on MRI;3). Presence of surgical implants.4). Presence of peripheral nerve compression. The interval between clinical symptom evaluation and MRI image review did not exceed one month for any patient.
6.Finally, it would be important to clarify that no external validation (i.e., patients from another hospital or different cohort) was employed in this study.
Reply: Thank you for your valuable comment. We have clarified in the limitations section that this study did not include external validation using data from other hospitals or independent cohorts. This acknowledgment highlights the need for future research to assess the model’s generalizability across diverse populations.
7.Where is the indicated Figure 2 (line 117-118)? In addition, it is placed before Figure 1.
- Tables 1 and 2 should also report all the metrics indicated in the text (e.g., F1-scores and AUCs).
- Table 3 is superfluous and should be eliminated (data should be included in the text).
- Figure 1 includes readers 3 and 4, it is not a confusion matrix, and it refers to
- Table 4 is not a confusion matrix but a contingency matrix of interobserver agreement. In addition, data sum equals 764 patients, thus it is related to the training/validation.
12.In general, the results section should be revised to report clear data in terms of both visual and textual information.
Reply: Thank you for your valuable comments. We have thoroughly revised all tables, figures, and the Results section to address the suggestions and improve clarity and coherence.
13.Line 74: do not forget neuroradiologists as experts apart from musculoskeletal radiologists.
Reply: Thank you for pointing out this important issue. Neuroradiologists has been added in manuscript.
14.The number of references used in the manuscript is not bad, but it could be enriched to reach 30, which seems more appropriate considering the extent of the manuscript. In the introduction, I would suggest to include some further information regarding how MRI allows to detect not only central canal spinal stenosis but also lateral recess and foraminal stenoses, as opposed to, see suggested reference:
Ruiz Santiago F, Láinez Ramos-Bossini AJ, Wáng YXJ, Martínez Barbero JP, García Espinosa J, Martínez Martínez A. The value of magnetic resonance imaging and computed tomography in the study of spinal disorders. Quant Imaging Med Surg. 2022 Jul;12(7):3947-3986. doi: 10.21037/qims-2022-04. PMID: 35782254; PMCID: PMC9246762.
Reply: Thank you for the helpful suggestion. We have incorporated the recommended reference by Ruiz Santiago et al., which provided valuable insights on MRI’s role in detecting not only central canal stenosis but also lateral recess and foraminal stenoses. Additionally, we have expanded the reference list to better reflect the scope of the manuscript.
15.Similarly, the following paper also explored a DL method to classify central canal stenosis at the cervical spine and -although not explicitly stated- they also applied the Kang’s and Park’s criteria, so this paper is not radically new (and the introduction/discussion should be modified accordingly):
Zhang E, Yao M, Li Y, Wang Q, Song X, Chen Y, Liu K, Zhao W, Xing X, Zhou Y, Meng F, Ouyang H, Chen G, Jiang L, Lang N, Jiang S, Yuan H. Deep learning model for the automated detection and classification of central canal and neural foraminal stenosis upon cervical spine magnetic resonance imaging. BMC Med Imaging. 2024 Nov 26;24(1):320. doi: 10.1186/s12880-024-01489-w. PMID: 39593012; PMCID: PMC11590449.
Reply: Thank you for highlighting this relevant study. We found the work by Zhang et al. very helpful and have accordingly revised our manuscript to better position our study’s novelty and to refine the conclusions in light of their findings.
16.Finally, English writing needs careful revision, particularly regarding the appropriate use of verb tenses; there are may sentences which inappropriately use present tenses instead of past tenses (e.g., lines 14, 17, and in several parts of the methodology section). There are also several typos (access instead of assess in line 7, precess instead of process in line 60), and the use of MR / MRI acronyms is inconsistent across the text. Please homogenize.
Reply: Thank you for your valuable feedback. We have thoroughly revised the manuscript to correct verb tense inconsistencies, fix typographical errors, and standardize the use of MR/MRI acronyms throughout the text, significantly improving the overall language quality.
Thank you for your consideration, and we look forward to hearing from you at your earliest convenience.
Sincerely yours,
Dr. Hai Lv

Round 2
Reviewer 1 Report
Comments and Suggestions for Authors
The authors have addressed the reviewer's suggestions, further improving the manuscript:
- Response 1 - Accepted - Clarified study goals and hypotheses
- Response 2 - Accepted - Justified clinical value over binary classification
- Response 3 - Accepted - Added recent literature citations (post-2021)
- Response 4 - Accepted - Improved statistical rigor and guideline adherence
- Response 5 - Accepted - Revised conclusions to better align with evidence
- Response 6 - Accepted - Added recent literature citations (post-2021)
- Response 7 - Accepted - Provided explanations for the absence of AI output figures
Author Response
May 28th, 2025
Dear Reviewer:
On behalf of all the contributing authors, I would like to express our sincere appreciations of your letter and reviewers' constructive comments concerning our article entitled “Deep Learning-Based Computer-Aided Grading of Cervical Spinal Stenosis from MR Images: Accuracy and Clinical Alignment” (Manuscript Number: bioengineering-3604451.). We appreciate the time and effort you have dedicated to evaluating our work. Thank you once again for your careful review and constructive guidance, which have been instrumental in strengthening our work. We truly appreciate your dedication and professionalism throughout this process. We hope this revised manuscript will meet with approval and be acceptable for publication in Bioengineering.
Sincerely yours,
Dr. Hai Lv
Reviewer 3 Report
Comments and Suggestions for Authors
Excellent explanation of my previously-expressed concerns. The amendments made have significantly improved the clarity and reading flow of the manuscript. I only miss a couple of lines in the Conclusion section in which the authors succinctly emphasize the potential clinical utility of their developed model, as well as the need to overcome the main limitations of the study. This would put a cherry on top of a well-designed and well-written paper.
Author Response
May 28th, 2025
Dear Reviewer:
On behalf of all the contributing authors, I would like to express our sincere appreciations of your letter and reviewers' constructive comments concerning our article entitled “Deep Learning-Based Computer-Aided Grading of Cervical Spinal Stenosis from MR Images: Accuracy and Clinical Alignment” (Manuscript Number: bioengineering-3604451.). We appreciate the time and effort you have dedicated to evaluating our work. We have carefully considered each of your suggestions and have made revisions to address the concerns raised. The corresponding changes were highlighted in red in the revised manuscript.
We are grateful for your positive assessment of the revisions and are pleased that the amendments have addressed the reviewers’ concerns and enhanced the manuscript. We fully agree with your final suggestion regarding the Conclusion section and changed it into:
“This study developed a deep learning-based intelligent Kang grading system for cervical spinal canal stenosis, which is supported by radiculopathy symptoms. The model was trained on a large dataset with relatively low inter-reader consistency from junior physicians and demonstrated improved grading performance on the test set, which can be beneficial to play the function of assisting imaging to clinician in clinical practical work, and also provide a reliable and convenient grading tool for further clinical high-quality studies related to the grading of cervical spinal stenosis in the future. It is recommended that the outputs of each individual model be carefully interpreted to effectively evaluate explainability. Moreover, future studies should address the principal limitations of this study to enhance the model’s generalizability and its potential for clinical translation.”
We hope this revised manuscript will meet with approval and be acceptable for publication in Bioengineering. Thank you once again for your careful review and constructive guidance, which have been instrumental in strengthening our work. We truly appreciate your dedication and professionalism throughout this process.
Sincerely yours,
Dr. Hai Lv